
**1  Biogenic and anthropogenic sources of isoprene and monoterpenes and their secondary organic**
**2  aerosol in Delhi, India**

Daniel J. Bryant[1], Beth S. Nelson[1], Stefan J. Swift[1a], Sri Hapsari Budisulistiorini[1], Will S. Drysdale[1,2],
Adam R. Vaughan[1], Mike J. Newland[1b], James R. Hopkins[1,2], James M. Cash[3,4], Ben Langford[3], Eiko
Nemitz[3], W. Joe F. Acton[5c], C. Nicholas Hewitt[5], Tuhin Mandal[6], Bhola R. Gurjar[6], Shivani[6d], Ranu
Gadi[6], James D. Lee[1,2], Andrew R. Rickard[1,2], Jacqueline F. Hamilton[1]
1- Wolfson Atmospheric Chemistry Laboratories, Department of Chemistry, University of York,
Heslington, York, YO10 5DD, UK
2- National Centre for Atmospheric Science, University of York, Heslington, York, YO10 5DD, UK
3- UK Centre for Ecology and Hydrology, Penicuik, Midlothian, Edinburgh, EH26 0QB, UK
4- School of Chemistry, University of Edinburgh, Edinburgh, EH9 3FJ, Edinburgh, UK
5- Lancaster Environment Centre, Lancaster University, Lancaster, LA1 4YW, UK
6- Department of Applied Sciences and Humanities, Indira Gandhi Delhi Technical University for
Women, Delhi, 110006, India
[a] now at: J. Heyrovsky Institute of Physical Chemistry, Department of Chemistry of Ions in Gaseous
Phase, Prague, Czech Republic
[b] now at: ICARE-CNRS, 1 C Av. de la Recherche Scientifique, 45071 Orléans CEDEX 2, France
[c] now at: School of Geography, Earth and Environmental Sciences, University of Birmingham,
Birmingham, B15 2TT, UK
[d] now at: Department of Chemistry, Miranda House, Delhi University, Delhi, 110007, India
Correspondence email: daniel.bryant@york.ac.uk

**22  Abstract**

Isoprene and monoterpenes emissions to the atmosphere are generally dominated by biogenic
sources. The oxidation of these compounds can lead to the production of secondary organic aerosol,
however the impact of this chemistry in polluted urban settings has been poorly studied.  Isoprene
and monoterpenes can form SOA heterogeneously via anthropogenic-biogenic interactions resulting
in the formation of organosulfates (OS) and nitrooxy-organosulfates (NOS). Delhi, India is one of the
most polluted cities in the world, but little is known about the emissions of biogenic VOCs or the
sources of SOA. As part of the DELHI-FLUX project, gas phase mixing ratios of isoprene and speciated
monoterpenes were measured during pre- and post-monsoon measurement campaigns in central
Delhi.  Nocturnal mixing ratios of the VOCs were substantially higher during the post-monsoon
(isoprene: $(0.65 \pm 0.43)$ ppbv, limonene: $(0.59 \pm 0.11)$ ppbv, $\alpha$-pinene: $(0.13 \pm 0.12)$ ppbv) than the
pre-monsoon (isoprene: $(0.13 \pm 0.18)$ ppbv, limonene: $0.011 \pm 0.025$ (ppbv), $\alpha$-pinene: $0.033 \pm$
$0.009$) period.  At night, isoprene and monoterpene concentrations correlated strongly with CO
across during the post-monsoon period. This is one of the first observations in Asia, suggesting
monoterpene emissions are dominated by anthropogenic sources. Filter samples of particulate
matter less than 2.5 microns in diameter ($PM_{2.5}$) were collected and the OS and NOS content
analysed using ultrahigh-performance liquid chromatography tandem mass spectrometry (UHPLC-
$MS^2$). Inorganic sulfate was shown to facilitate the formation of isoprene OS species across both
campaigns. Sulfate contained within OS and NOS species were shown to contribute significantly to



the sulfate signal measured via AMS. Strong nocturnal enhancements of NOS species were observed
across both campaigns. The total concentration of OS/NOS species contributed an average of (2.0 ±
0.9) % and (1.8 ± 1.4) % to the total oxidised organic aerosol, and up to a maximum of 4.2 % and 6.6
% across the pre- and post-monsoon periods, respectively. Overall, this study provides the first
molecular level measurements of SOA derived from isoprene and monoterpene in Delhi and
demonstrates that both biogenic and anthropogenic sources of these compounds can be important
in urban areas.
**1. Introduction**
India is undergoing significant urbanization and industrialisation, with a rapidly increasing
population. According to the WHO, India was home to 9 out of the top 10 most polluted cities in the
world in 2020 in terms of annual mean $PM_{2.5}$ (particulate matter less than 2.5 micrometres in
diameter) concentrations (WHO, 2018). In Delhi, the population-weighted mean $PM_{2.5}$ was estimated
to be 209 (range: 120 – 339.5) $\mu g\ m^{-3}$ in 2017, over 40 times the WHO annual mean guidelines of 5
$\mu g\ m^{-3}$, and greater than five times India's own standard of 40 $\mu g\ m^{-3}$ (Balakrishnan et al., 2019). Air
pollution is estimated to cause over 1 million deaths per year in India alone (Landrigan et al., 2018).
Numerous studies have investigated $PM_{2.5}$ concentrations, characteristics and meteorological effects
in Delhi (Anand et al., 2019; Bhandari et al., 2020; Chowdhury et al., 2004; Hama et al., 2020;
Kanawade et al., 2020; Miyazaki et al., 2009; Nagar et al., 2017). The key sources of $PM_{2.5}$ identified
are secondary aerosol, fossil fuel combustion, municipal waste and biomass burning (Chowdhury et
al., 2004; Sharma and Mandal, 2017; Stewart et al., 2021b, 2021a). Previous studies have also shown
that alongside extremely high emissions of pollutants, regional sources and meteorology in
particular play an important role in high pollution events in Delhi (Bhandari et al., 2020; Sawlani et
al., 2019; Schnell et al., 2018; Sinha et al., 2014).
Secondary species have been shown to be significant contributors to $PM_1$ and $PM_{2.5}$ mass in Delhi,
with organics contributing 40-70 % of $PM_1$ mass. (Gani et al., 2019; Shivani et al., 2019; Reyes-
Villegas et al., 2021; Sharma and Mandal, 2017) However, limited molecular level analysis of organic
aerosol (OA) has been undertaken (Chowdhury et al., 2004; Elzein et al., 2020; Miyazaki et al., 2009;
Singh et al., 2021, 2012; Yadav et al., 2021). Kirillova et al., (2014) analysed the sources of water-
soluble organic carbon (WSOC) in Delhi, using radiocarbon measurement constraints. The study
identified that 79 % of WSOC was classified as non-fossil carbon, attributed to biogenic/biomass
burning sources in urban Delhi (Kirillova et al., 2014), similar to other studies from India (Kirillova et
al., 2013; Sheesley et al., 2012). Studies across Asia, Europe and North America have also shown high
contributions from non-fossil sources to ambient PM concentrations in urban environments (Du et
al., 2014; Kirillova et al., 2010; Szidat et al., 2004; Wozniak et al., 2012). The sources of this modern
carbon in urban areas are poorly understood, although biomass burning is a key component (Elser et
al., 2016; Hu et al., 2016; Lanz et al., 2010; Nagar et al., 2017). Recently in Delhi, solid-fuel
combustion sources such as cow dung cake or municipal solid waste have been shown to release
over 1000 different organic components into the aerosol phase at emission (Stewart et al., 2021a).
Alongside biomass burning, one potential source of this non-fossil aerosol is biogenic secondary
organic aerosol (BSOA), which is formed via the oxidation of biogenic volatile organic compounds
(BVOCs) and subsequent gas-particle phase transfer (Hallquist et al., 2009; Hoffmann et al., 1997).
Isoprene is the most abundant BVOC, with annual global emissions estimates of between 350 - 800
$Tg\ yr^{-1}$ (Guenther et al., 2012; Sindelarova et al., 2014). Globally, isoprene is predominately emitted
from biogenic sources, but anthropogenic sources become increasingly important in urban areas
especially at night (Borbon et al., 2001; Hsieh et al., 2017; Khan et al., 2018a; Mishra and Sinha,





2020; Sahu et al., 2017; Sahu and Saxena, 2015). Monoterpenes are another important BSOA
precursor, with annual global emissions estimates of between 89 and 177 Tg yr$^{-1}$ (Guenther et al.,
2012; Sindelarova et al., 2014). Monoterpenes while mainly biogenic, are also emitted from
anthropogenic sources such as biomass burning, cooking and fragranced consumer products (Cheng
et al., 2018; Gkatzelis et al., 2021; Panopoulou et al., 2020, 2021; Stewart et al., 2021b, 2021c; Zhang
et al., 2020).
Numerous studies have identified and quantified molecular level markers from isoprene and
monoterpenes, especially in the Southeastern-US and China (Brüggemann et al., 2019; Bryant et al.,
2020, 2021; Hettiyadura et al., 2019; Huang et al., 2016; Rattanavaraha et al., 2016b; Wang et al.,
2016, 2018a; Yee et al., 2020).  The complex sources of isoprene and monoterpenes in highly
polluted urban areas make source identification difficult. As such, the SOA markers in this study will
be referred to as originating from isoprene or monoterpenes, but the emissions are likely from a
mixture of biogenic and anthropogenic sources as discussed previously. (Cash et al., 2021b; Nelson
et al., 2021)
Recent studies have started to focus on anthropogenic-biogenic interactions, whereby
anthropogenic pollutants such as NO$_x$ and sulfate enhance the formation of biogenically derived SOA
species. Increased NO or NO$_2$ concentrations can lead to higher organonitrate (ON) or nitrooxy-
organosulfate (NOS) concentrations through RO$_2$ + NO or VOC + NO$_3$ pathways.(Morales et al., 2021;
Takeuchi and Ng, 2019) Inorganic sulfate formed from the oxidation of SO$_2$ plays a pivotal role in OS
and NOS formation (Bryant et al., 2020; Budisulistiorini et al., 2015; Glasius et al., 2018; Hettiyadura
et al., 2019; Hoyle et al., 2011; Xu et al., 2015). Sulfate allows the acid-catalysed uptake of gas phase
oxidation products into the particle phase. Both chamber and ambient studies have shown the direct
link between sulfate and OS concentrations (Brüggemann et al., 2020a; Bryant et al., 2020;
Budisulistiorini et al., 2015; Gaston et al., 2014; Lin et al., 2012; Riva et al., 2019; Surratt et al.,
2008a; Xu et al., 2015). Yee et al., (2020) highlighted markers from both the high/low-NO isoprene
oxidation pathways correlated linearly with sulfate over a large range of sulfate concentrations (0.01
– 10 µg m$^{-3}$) across central Amazonia during the wet and dry seasons and in the SE-US summer. They
conclude that the majority of isoprene oxidation products in pre-industrial settings are still expected
to be in the form of isoprene OS (OSi), suggesting that they cannot be thought of as purely a
biogenic-anthropogenic product (Yee et al., 2020).
In this study, offline PM$_{2.5}$ filter samples were collected across two campaigns (pre and post-
monsoon) in central Delhi, alongside a comprehensive suite of gas and aerosol atmospheric
pollutant measurements. Filters were analysed using ultra-high performance liquid chromatography
tandem mass spectrometry and isoprene and monoterpene OS/NOS markers identified and
quantified. Isoprene and monoterpene emissions were observed to correlate strongly to
anthropogenic markers, suggesting a mixed anthropogenic/biogenic sources of these VOCs. OSi
species showed strong seasonality and strong correlations to particulate sulfate. NOS species
showed strong nocturnal enhancements, likely due to nitrate radical chemistry. This study is the first
molecular level particle phase analysis of OS and NOS markers from isoprene and monoterpenes in
Delhi and aims to improve our understanding of the sources of isoprene and monoterpene SOA
markers and their formation pathways in extremely polluted urban environments.
**2.Experimental**
**2.1 Filter collection and site information**


PM$_{2.5}$ filter samples were collected as part of the Air Pollution and Human Health (APHH)-India
campaign, at the Indira Gandhi Delhi Technical University for Women in New Delhi, India, (28°39'55"
N 77°13'56" E). The site is situated inside the third ring road which caters to huge volumes of traffic,
with a major road to the east, between the site and the Yamuna River. Two train stations are located
to the south and southwest of the site, and there are several green spaces locally in all
directions.(Nelson et al., 2021; Stewart et al., 2021c) Filters were collected during two field
campaigns in 2018. The first campaign was during the pre-monsoon period, with 35 filters were
collected between 28/05/2018 and 05/06/2018. The second campaign during the post-monsoon
period, 108 filters were collected between 09/10/2018 and 6/11/2018. Quartz filters (Whatman
QMA, 10" by 8") were pre-baked at 550 °C for 5 hours and wrapped in foil before use. Samples were
collected using an HiVol sampler (Ecotech 3000, Victoria Australia) with selective PM$_{2.5}$ inlet at a flow
rate of 1.33 m$^3$ min$^{-1}$. Once collected, filters were stored in foil at -20 °C before, during and after
transport for UK based analysis.
**2.2 Filter extraction**
Using a standard square filter cutter, a section of filter was taken with an area of 30.25 cm$^2$ which
was then cut into roughly 1 cm$^2$ pieces and placed in a 20 mL glass vial. Next, 8 mL of LC-MS grade
methanol (MeOH, Optima, Fisher Chemical, USA) was added to the sample and sonicated for 45 min.
Ice packs were used to keep the bath temperature below room temperature, with the water
swapped mid-way through. Using a 5 mL plastic syringe, the MeOH extract was then pushed through
a 0.22 µm filter (Millipore) into another sample vial. An additional 2 mL (2 x 1 mL) of MeOH was
added to the filter sample, and then extracted through the filter to give a combined extract ~ 10mL.
This extract was then reduced to dryness using a Genevac solvent evaporator under vacuum. The dry
sample was then reconstituted in 50:50 MeOH:H$_2$O (Optima, Fisher Chemical, USA) for analysis
(Bryant et al., 2020; Spolnik et al., 2018). Extraction efficiencies of 2-methyl-glyceric acid (2-MG-OS)
and camphorsulfonic acid were determined using authentic standards spiked onto a pre-baked clean
filter and recoveries were calculated to be 71 % and 99 % respectively.
**2.3 Ultra-high performance liquid chromatography tandem mass spectrometry (UHPLC-MS$^2$)**
The extracted fractions of the filter samples were analysed using an Ultimate 3000 UHPLC (Thermo
Scientific, USA) coupled to a Q-Exactive Orbitrap MS (Thermo Fisher Scientific, USA) using data
dependent tandem mass spectrometry (ddMS$^2$) with heated electrospray ionization source (HESI).
The UHPLC method uses a reversed-phase 5 µm, 4.6 mm × 100 mm, polar end capped Accucore
column (Thermo Scientific, UK) held at 40 °C. The mobile phase consisted of water (A, optima grade)
and methanol (B, optima grade) both with 0.1 % (v/v) of formic acid (98 % purity, Acros Organics).
Gradient elution was used, starting at 90 % (A) with a 1-minute post-injection hold, decreasing to 10
% (A) at 26 minutes, returning to the starting mobile phase conditions at 28 minutes, followed by a
2-minute hold allowing the re-equilibration of the column. The flow rate was set to 0.3 mL min$^{-1}$. A
sample injection volume of 4 µL was used. The capillary and auxiliary gas heater temperatures were
set to 320 °C, with a sheath gas flow rate of 45 (arb.) and an auxiliary gas flow rate of 20 (arb.).
Spectra were acquired in the negative ionization mode with a scan range of mass-to-charge (*m/z*) 50
to 750. Tandem mass spectrometry was performed using higher-energy collision dissociation with a
stepped normalized collision energy of 10,45 and 60. The isolation window was set to *m/z* 2.0 with a
loop count of 10, selecting the 10 most abundant species for fragmentation in each scan.
A mass spectral library was built using the compound database function in Tracefinder 4.1 General
Quan software (Thermo Fisher Scientific, USA). To build the library, compounds from previous
studies (Chan et al., 2010; Nestorowicz et al., 2018; Ng et al., 2008; Riva et al., 2016b; Schindelka et



al., 2013; Surratt et al., 2008a) were searched for in an afternoon and a night-time filter sample
extract analysis using the Xcalibur software. Further details can be found in Bryant et al., 2021 and
the SI. Isoprene and monoterpene markers were quantified using the method in Bryant et al., 2021.
Overall uncertainties associated with calibrations, proxy standards and matrix effects were
estimated. The uncertainties associated with 2-MG-OS and 2-methyl tetrol OS (2-MT-OS) were
calculated to be 58.9 % and 37.6 % respectively, mainly due to the large uncertainties in the matrix
correction factors. Isoprene SOA markers quantified by the average of 2-MT-OS and 2-MG-OS
calibrations have an associated uncertainty of 69.9 %. For monoterpene SOA species which were
quantified by camphorsulfonic acid, the associated uncertainty is estimated to be 24.8 %.

**2.4 Supplementary measurements**
A suite of complementary measurements were made alongside the filter collection including
VOCs(Stewart et al., 2021c), oxygenated-VOCs, $NO_x$, CO, $O_3$, $SO_2$, HONO, photolysis rates and
measurements of $PM_1$ non-refractory aerosol chemical components with a high resolution Aerosol
Mass Spectrometer (HR-AMS). Detailed instrument descriptions can be found in Nelson et al.,
(2021). Briefly, VOCs and oxygenated-VOCs were measured via two gas-chromatography (GC)
instruments (DC-GS-FID and GC-GC-FID). $NO_x$ was measured via a dual channel chemiluminescence
analyser with fitted with a blue light converter for $NO_2$ (Air Quality Designs Inc., Colorado) alongside
CO which was measured with a resonance fluorescent instrument (Model Al5002, Aerolaser GmbH,
Germany). $O_3$ was measured as outlined by Squires et al., (2020) using an ozone analyser (49i,
Thermo Scientific). $SO_2$ was measured using a 43i $SO_2$ analyser (Thermo scientific).  High-resolution
aerosol mass spectrometry measurements were conducted as outlined in Cash et al., (2021). Ion
chromatography measurements were undertaken by the experimental approach outlined by Xu et
al., (2020) as part of an intercomparison study. Briefly, filter cuttings were taken from the filter and
extracted ultrasonically for 30 mins in 10 mL of ultrapure water and then filtered before analysis (Xu
et al., 2020).
Meteorology data was downloaded from the NOAA Integrated Surface Database via the Worldmet R
package for the Indira Gandhi International Airport (code: 421810-99999) (Carslaw, D ., accessed:
2021). The planetary boundary layer height (PBLH) was obtained from the ERA5 (ECMWF ReAnalysis
5) data product at 0.25° resolution in 1-hour time steps at the position Lat 28.625˚, Lon. 77.25˚. The
data for both campaigns was then selected between the start time of the first filter of that
campaign, and the end time of the last filter of the same campaign.
**3. Results**
**3.1 Meteorology**
The time series for temperature, RH, planetary boundary layer height (PBLH) and ventilation
coefficient (VC) across the pre- and post-monsoon campaigns are shown in Figure S1. For the pre-
monsoon campaign, the average air temperature was (35.8 ± 4.5) °C compared to (24.7 ± 4.6) °C in
the post-monsoon campaign (Table S2). The pre-monsoon campaign also showed higher average
wind speeds, with an average of (3.8 ± 1.4) ms⁻¹, compared to (1.7 ± 1.3) ms⁻¹ in the post-monsoon
campaign. The average RH of the pre- and post-monsoon were (39.4 ± 13.6) % and (57.3 ± 16.6) %
respectively, both showing similar diurnals with a minimum around mid-morning and nocturnal
maximum (Figure S2). The PBLH shows a similar diurnal between the two campaigns, with the
nocturnal boundary layer breaking down around 06:00-07:00 with a midday peak, before re-
establishing the nocturnal boundary layer around 19:00. The pre-monsoon PBLH has an average



maximum of ~2400 m compared to post-monsoon ~1700 m and a minimum of 270 m compared to
52 m (Figure S2). The ventilation coefficient (VC = wind speed x PBLH) has been used previously to
identify periods of adverse meteorological conditions and gives an idea of how stagnant atmospheric
conditions are and the general role of the atmosphere in the dilution of species. (Gani et al., 2019)
As shown in Figure S1, the conditions during the post-monsoon campaign were much more stagnant
than the pre-monsoon campaign. The VC was on average 4.5 times higher during the pre-monsoon
campaign compared to the post-monsoon campaign (Table S2) in line with previous studies (Gani et
al., 2019; Saha et al., 2019). The more stagnant conditions during the post-monsoon campaign likely
traps nocturnal emissions and their reaction products close to the surface, allowing for a significant
build-up of concentrations.
**3.2 Gas phase observations**
Time series of the observed mixing ratios (ppbv) of NO, $NO_2$ and $O_3$ are shown in Figure 1, for the
pre- and post-monsoon campaigns. The campaign averaged diurnal profiles are shown in Figure S3
and the mean, median and maximum mixing ratios are given in Table S2. It should be noted that only
one week of data was available for the pre-monsoon period. During the post-monsoon campaign,
extremely high mixing ratios of NO were observed with a campaign maximum mixing ratio of ~870
ppbv during the early morning of the 1st of November. During the early part of the pre-monsoon
campaign, a large enhancement in NO was observed with mixing ratios around 400 ppbv (Figure S4),
followed by significantly lower concentrations throughout the rest of the campaign. The campaign-
average NO diurnal profile shows very high NO mixing ratios at night (pre-: ~ 50 ppbv, post-: ~300
ppbv), with low afternoon mixing ratios < 2 ppbv due to ozone titration. These high NO
concentrations at night likely reduce any night-time chemistry through reactions with $NO_3$ radicals
and ozone. $NO_2$ during the pre-monsoon was observed to increase as the boundary layer reduced in
the late afternoon, with a mid-afternoon minimum. During the post-monsoon, a double peak in
concentrations was observed, in line with increasing ozone in the morning, and increasing NO in the
afternoon. Ozone showed a strong diurnal variation across both campaigns, with average afternoon
mixing ratios ~ 75 ppbv with pre- and post-monsoon maximums of 182 ppbv and 134 ppbv
respectively. Night-time $O_3$ concentrations were significantly higher during the pre-monsoon
campaign, likely due to the significantly lower NO concentrations.
**3.3 Particle phase observations**
The sampling site was heavily polluted in terms of particulate matter. The mean ± σ $PM_{2.5}$
concentration (Table S2) during the pre-monsoon campaign was (141 ± 31) µg m$^{-3}$ with a spike in
concentrations of 672 µg m$^{-3}$ on the 01/6/2018 at 21:00 (Figure 1). The diurnal (Figure S5) shows
concentrations generally flat throughout the day. During the post-monsoon campaign, the average
$PM_{2.5}$ concentration was higher at (182 ± 94) µg m$^{-3}$, with a spike in concentrations of 695 µg m$^{-3}$ at
the end of the campaign (Figure 1). The diurnal shows a mid-afternoon minimum with high morning
and night concentrations. HR-AMS was used to measure the $PM_1$ sulfate and total organics.
Campaign averaged total organics concentrations were approximately double in the post-monsoon
(48.7 ± 35.4) µg m$^{-3}$ compared to the pre-monsoon (19.8 ± 13.7) µg m$^{-3}$. During the pre-monsoon
campaign, concentrations are generally flat throughout the day, with an increase in the late
afternoon, likely as the boundary layer decreases (Figure S5). During the post-monsoon, a much
more prominent diurnal is observed, with a mid-day minimum and high night-time concentrations.
This diurnal is likely driven by boundary layer conditions. Sulfate averaged (7.5 ± 1.8) µg m$^{-3}$ during
the pre-monsoon campaign, with slightly lower average concentrations observed in the post-
monsoon: (5.6 ± 2.7) µg m$^{-3}$ as shown in Figure S5. The sulfate diurnal variations are similar to those
of the organic aerosol.

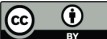





















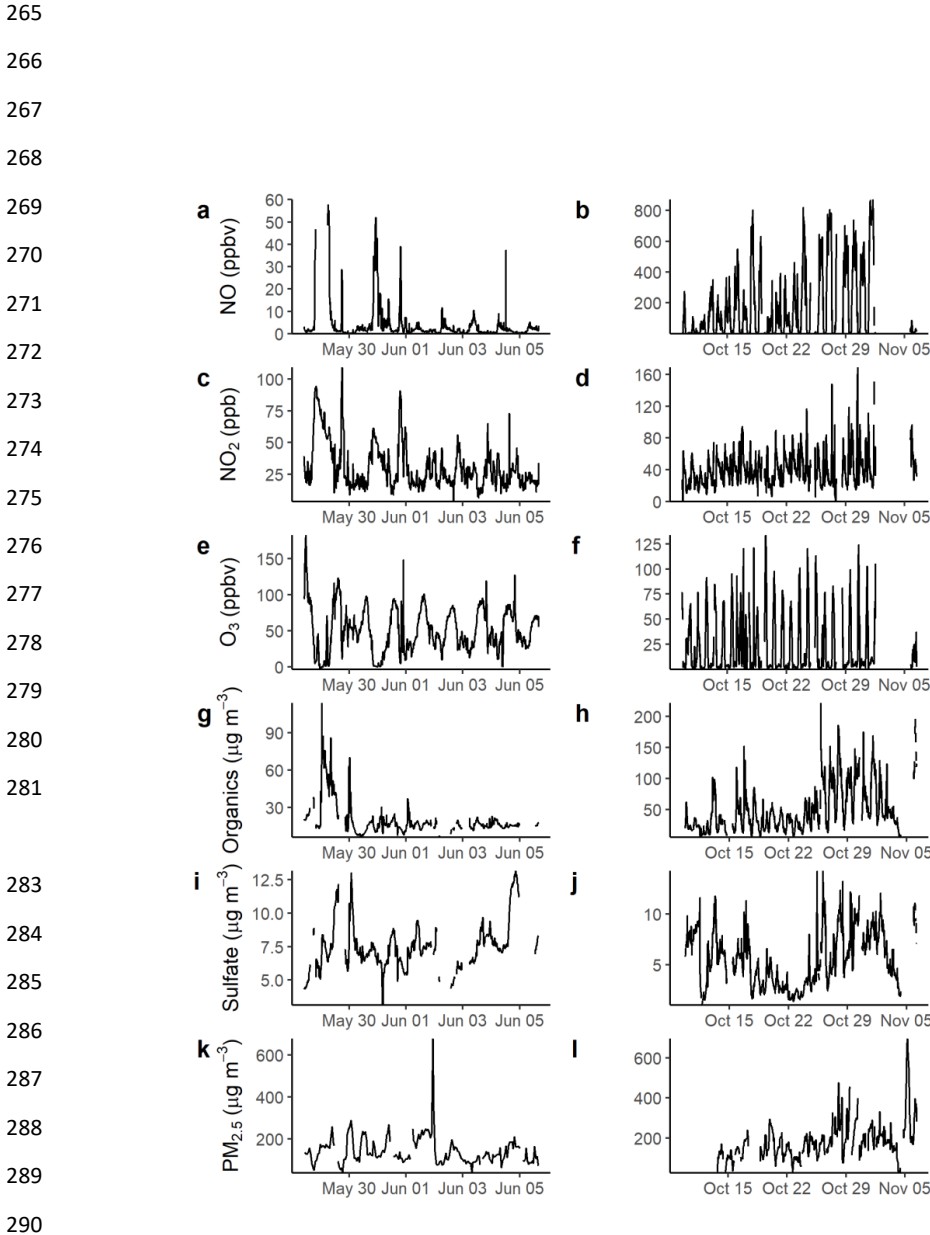









Figure 1. Time series of pollutants across the pre- (a,c,e,g,i) and post-monsoon (b,d,f,h,j)
campaigns. During the pre-monsoon, NO concentrations were filters to below 60 ppbv, due to
       a large enhancement in concentrations at the start of the campaign, the full time series is
shown in Figure S4. NO, NO₂, O₃ and HR-AMS – SO₄²⁻ were averaged to 15 minutes. PM₂.₅ was
       measured hourly.






**3.4 Isoprene and monoterpene measurements**
Isoprene was measured hourly using gas-chromatography with flame-ionisation-detection (GC-FID)
across the two campaigns (Nelson et al., 2021), with the time series shown in Figure 2. The time
series highlights similar diurnal variability each day, driven by biogenic emissions. Figure 3 shows the
average diurnal profiles of isoprene during pre-monsoon (a) and post-monsoon (b). The mean
isoprene mixing ratios were (1.22 ± 1.28) ppbv and (0.93 ± 0.65) ppbv, with maximum isoprene
mixing ratios of 4.6 ppbv and 6.6 ppbv across the pre- and post-monsoon, respectively. This is in the
same range as measured in Beijing (winter mean: (1.21 ± 1.03) ppbv, summer mean: (0.56 ± 0.55)
ppbv, Acton et al., (2020)), Guangzhou (year round (1.14) ppbv) (Zou et al., 2019) and Taipei
(summer daytime: (1.26) ppbv, autumn daytime: (0.38) ppbv) (Wang et al., 2013). The diurnal
variability observed in the pre-monsoon period corresponds to a typical biogenic emission driven
profile, with a rapid increase of isoprene around 05:00, reaching a peak around or after midday,
before a nocturnal minimum. Figure 3 indicates that average daytime peak isoprene mixing ratios
during the pre-monsoon campaign were roughly double that of the post-monsoon campaign. In
contrast, average nocturnal mixing ratios of isoprene were 5 times higher in the post-monsoon
compared to the pre-monsoon ((0.65 ± 0.43) ppbv versus (0.13 ± 0.18) ppbv). In the post-monsoon
campaign, isoprene mixing ratios show a strong biogenic emission driven diurnal profile at the start
of the campaign. However, towards the end of the post monsoon measurement period, the isoprene
mixing ratios become less variable with a high mixing ratio maintained overnight (Figure 2). This is
potentially due to more stagnant conditions as observed by the VC in Figure S1.
















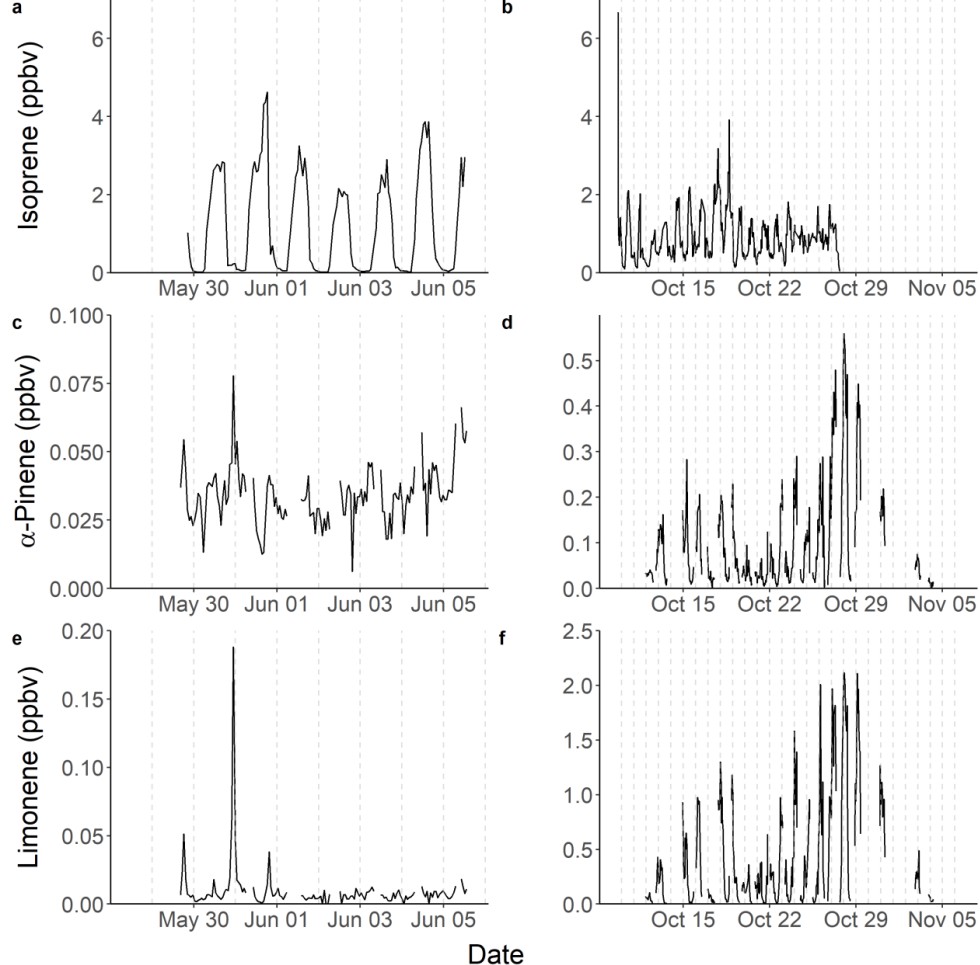


Figure 2. Time series across the pre- (left) and post-monsoon (right) campaigns of
Isoprene (a,b), α-pinene (c,d), limonene (d,e). The vertical dotted lines represent midnight
for each day.

A recent study in Delhi averaged across post-monsoon, summer and winter campaigns found that at
vegetative sites biogenic isoprene contributed on average 92 - 96 % to the total isoprene, while at
traffic dominated sites only 30 − 39 % of isoprene was from biogenic sources (Kashyap et al., 2019).
This is similar to the contributions of biogenic isoprene (40 %) to total isoprene mixing ratios at the
traffic dominated Marylebone Road London site.(Khan et al., 2018a) To gain some understanding of
the sources of isoprene at our site in Delhi, the observed concentrations of isoprene were correlated
to CO, which is an anthropogenic combustion tracer (Figure 4) similar to previous studies.(Khan et





al., 2018a; Wagner and Kuttler, 2014) The isoprene concentrations were split between night and day
(pre-monsoon; night: 19:00 – 05:00 h, day 05:00 – 19:00 h, post-monsoon; night: 17:00-06:00 h, day:
06:00-17:00 h), based on the observed isoprene diurnals as shown in Figure 3. Isoprene correlated
strongly with CO during the night across both campaigns (pre-monsoon: $R^2$= 0.69, post-monsoon:
$R^2$= 0.81), but no correlation was observed during the day ($R^2 < 0.1$). This suggests that daytime
isoprene is predominantly from biogenic sources, although a small amount will be from
anthropogenic sources, and that nocturnal isoprene is emitted from anthropogenic sources, as seen
in other locations. (Khan et al., 2018b; Panopoulou et al., 2020; Wang et al., 2013) The night-time
isoprene mixing ratios (pre-monsoon: 0.13 ± 0.18 ppbv, post-monsoon: 0.65 ± 0.43 ppbv) were
substantially higher than measured previously in Beijing and London ( <50pptv, (Bryant et al., 2020;
Khan et al., 2018b)), but pre-monsoon concentrations were similar to those observed at night in
Taipei (0.19 ppbv)(Wang et al., 2013). The high night-time concentrations during the post-monsoon
period, towards the end of October are also likely influenced by the formation of a very low
boundary layer, trapping pollutants near the surface, affecting all species similarly. An increase in
biomass burning may also be a factor. Therefore, during the post-monsoon campaign a significant
amount of isoprene oxidation products will be of anthropogenic origin.

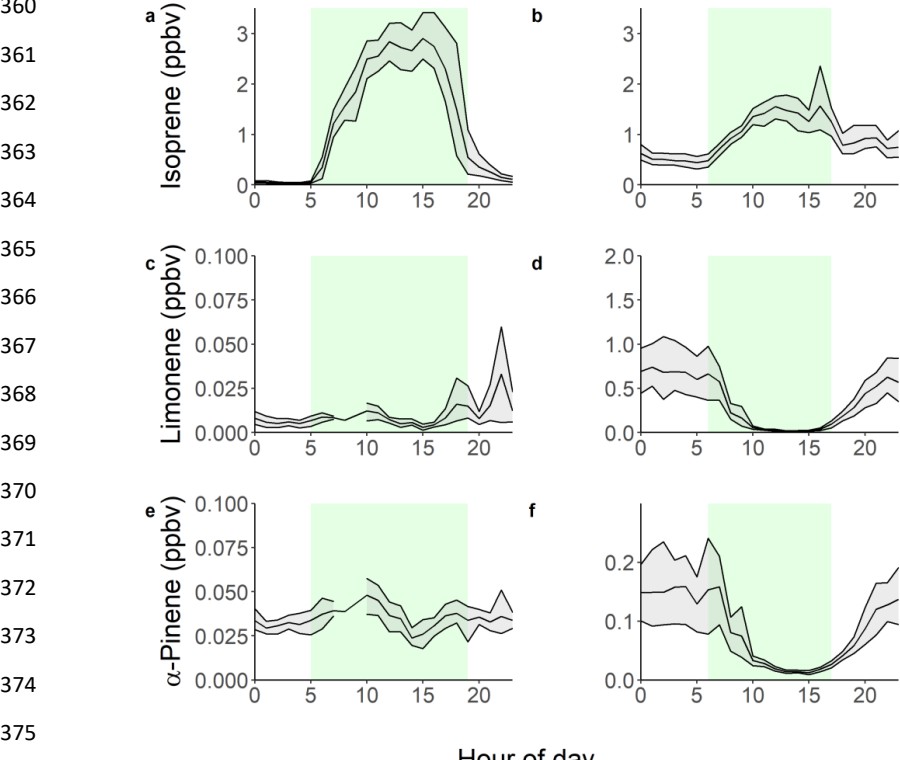

Hour of day

Figure 3. Diurnal variations across the pre (left) and post-monsoon (right) campaigns of Isoprene (a,b), limonene (c,d) and α-pinene (e,f). The grey shaded area represents the 95 % confidence interval. The green shaded area represents the times driven by biogenic emissions, as defined by the isoprene diurnals.




Several monoterpenes were measured using GCxGC-MS. The time series of two monoterpenes,
limonene and α-pinene, are shown in Figure 2. The α-pinene mixing ratio averaged (0.034 ± 0.011)
ppbv during the pre-monsoon and (0.10 ± 0.11) ppbv during the post monsoon periods. This is in
comparison to limonene, which averaged (0.01 ± 0.02) ppbv and (0.42 ± 0.51) ppbv across the pre-
and post-monsoon campaigns, respectively.  A strong diurnal variation was observed for both
monoterpenes during the post-monsoon, peaking during the night (Figure 3), with a midday
minimum. Nocturnal mixing ratios of the two monoterpenes were substantially higher during the
post-monsoon (Limonene: (0.59 ± 0.11) ppbv, α-pinene: (0.13 ± 0.12) ppbv) than the pre-monsoon
(Limonene: (0.011 ± 0.025) ppbv, α-pinene: (0.033 ± 0.009) ppbv) period. This diurnal again is likely
driven by boundary layer dynamics. During the pre-monsoon, limited diurnal variability was
observed compared to the post-monsoon. Limonene was dominated by 3 short lived spikes in
concentrations towards the start of the campaign (Figure 2). α-pinene concentrations generally
increased during the morning, before decreasing during the afternoon. A further 10 monoterpenes
were measured concurrently using GCxGC-MS (Nelson et al., 2021; Stewart et al., 2021c). For all MT
species, the post monsoon period had higher mean mixing ratios, with large nocturnal
enhancements in mixing ratios.
During the post-monsoon, α-pinene and limonene correlated strongly with CO during the day (α-
pinene; $R^2$ = 0.82, limonene; $R^2$ = 0.90) and moderately at night (α-pinene; $R^2$ = 0.49, limonene; $R^2$ =
0.56) as shown in Figure 4, suggesting anthropogenic sources. Other potentially important
anthropogenic monoterpene sources include biomass burning, cooking and the use of personal
care/volatile chemical products (Coggon et al., 2018; Gkatzelis et al., 2021; Hatch et al., 2019; Klein
et al., 2016). The shallow nocturnal boundary layers across both campaigns leads to relatively high
concentrations of total monoterpenes, with a maximum mixing ratio of 6 ppbv observed during the
post-monsoon (Stewart et al., 2021c). After sunrise, the expanding boundary layer dilutes the high
concentrations alongside increasing OH concentrations from photolytic sources such as the
photolysis of HONO and carbonyls which likely causes a rapid decrease in the monoterpene mixing
ratios.













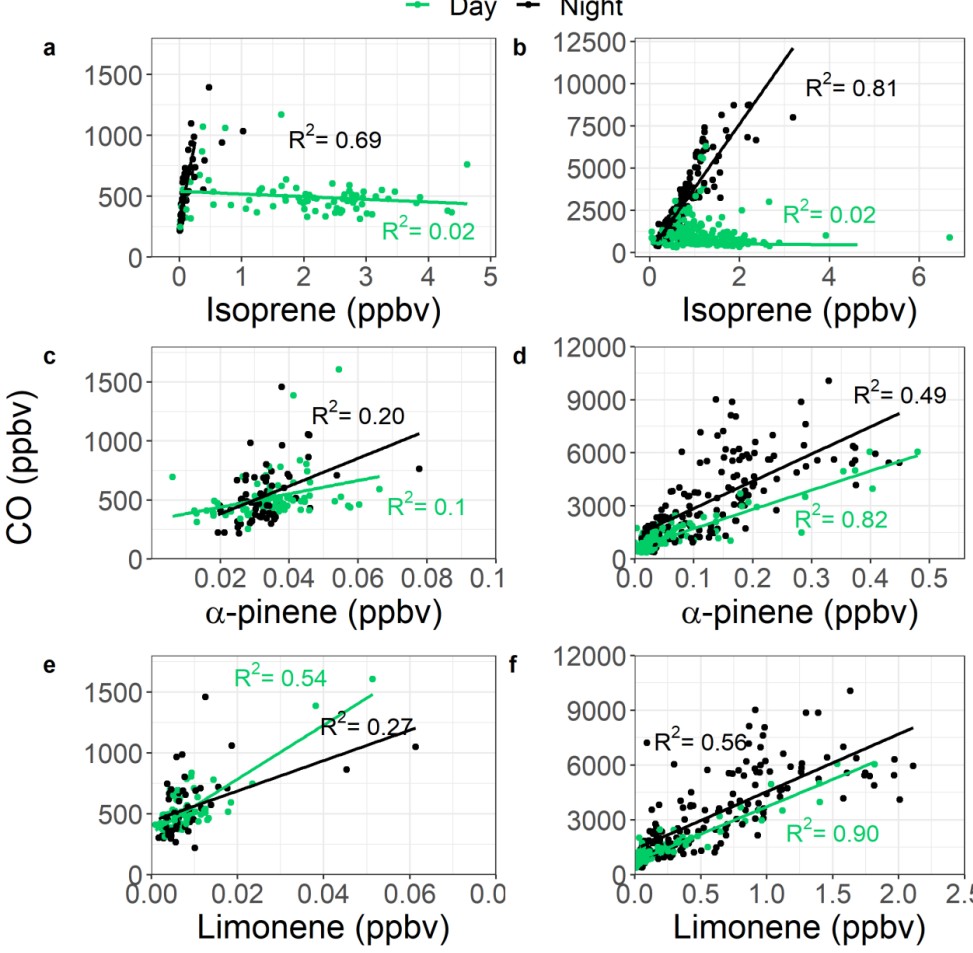


Figure 4. Correlations between Isoprene, limonene and α-pinene with CO across the pre (left)
and post-monsoon (right) campaigns. The samples are split between daytime (green) and night-
     time (black) as defined by the Isoprene diurnals in Figure 3.









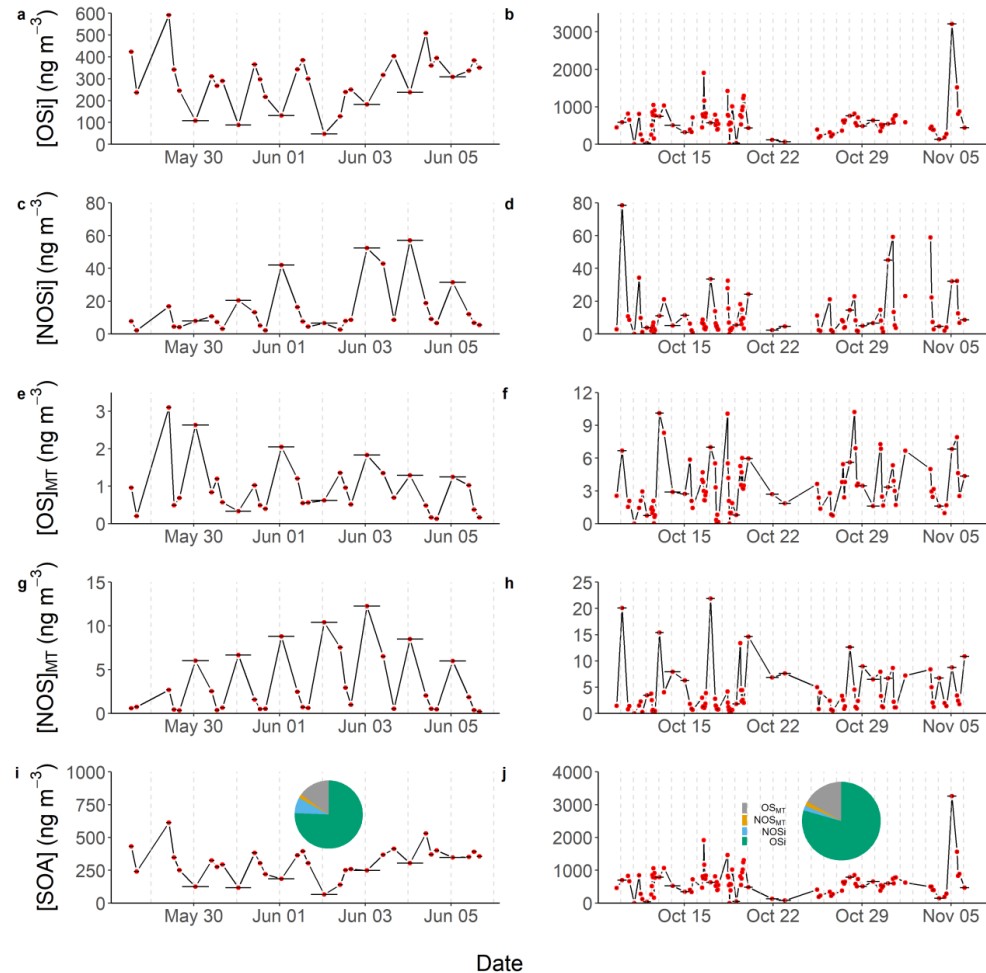


Figure 5. Time series across the pre- (left) and post-monsoon (right) campaigns of the quantified SOA tracers: OSi (a,b), NOSi (c,d), $OS_{MT}$ (e,f), $NOS_{MT}$ (g,h) and the sum of all SOA tarcers (i,j) with the average campaign contributions. The vertical dotted lines represent midnight for each day. Only species identified in more than 40 % of the samples for each campaign were included.


**3.5 Secondary organic aerosol formation**



At the measured concentrations, monoterpenes and isoprene are an important source of ozone and
OH reactivity at this site (Nelson et al., 2021). The resultant oxidised products will also be a key
source of SOA production. The UHPLC-MS$^2$ analysis identified and quantified 75 potential markers
across four classes of SOA, isoprene OS (OSi) and NOS (NOSi) derived species and monoterpene OS
(OS$_{MT}$) and NOS (NOS$_{MT}$) species. Figure 5 shows the contribution to the total quantified SOA (qSOA),
which consists of the time averaged sum of the four SOA classes (OSi, NOSi, OS$_{MT}$, NOS$_{MT}$), across the
pre- and post-monsoon campaigns. OSi species were the dominant SOA class quantified in this
study, contributing 75.6 % and 79.4 % of the qSOA across the pre- and post-monsoon campaigns
respectively. NOSi species contributed significantly more to the qSOA during the pre-monsoon (7.6
%) compared to the post-monsoon (2.1 %) period. Similar contributions from the monoterpene
derived SOA species were observed across both campaigns.

### 452    3.5.1 Isoprene SOA

OSi species are predominantly formed by photo-oxidation of isoprene by OH radicals with the
subsequent products formed dependent on ambient NO concentrations (Wennberg et al., 2018).
The pathways are split into high-NO and low-NO, although the NO concentrations that constitute
high and low are a sliding scale depending on the amount of reactivity (defined as ([OH] x $k_{OH}$)
(Newland et al., 2021). Under low-NO conditions, isoprene epoxydiol isomers (IEPOX) (Paulot et al.,
2009) are formed which can then undergo reactive uptake to the particle phase by acid-catalysed
multiphase chemistry involving inorganic sulfate, to form 2-MT-OS (Lin et al., 2012; Riva et al., 2019;
Surratt et al., 2010). Under high-NO conditions, 2-methyl glyceric acid is the dominant gas-phase
marker produced, which can undergo reactive uptake to the particle phase to form 2-MG-OS (Lin et
al., 2013a; Nguyen et al., 2015; Surratt et al., 2006, 2010).
A total of 21 potential OSi C$_{2-5}$ markers previously identified in chamber studies (Nguyen et al., 2010;
Riva et al., 2016a; Surratt et al., 2007, 2008b) and other ambient studies (Bryant et al., 2020;
Budisulistiorini et al., 2015; Hettiyadura et al., 2019; Kourtchev et al., 2016; Rattanavaraha et al.,
2016a; Wang et al., 2018b, 2021b) were quantified in the collected ambient samples. It should be
noted that several of the smaller (C$_{2-3}$) OSi tracers likely form from glyoxal, methylglyoxal and
hydroxyacetone as well as isoprene, and as such present a potential non-isoprene source of OSi
(Galloway et al., 2009; Liao et al., 2015).
Figure 5 shows the time series of total OSi concentrations observed across pre- (left, 5a) and post-
(right, 5b) monsoon campaigns. Total OSi time averaged concentrations (Table 1) were c.a. 2.3 times
higher during the post-monsoon (~556.6 ± 422.5 ng m$^{-3}$) campaign than the pre-monsoon campaign
(~237.8 ± 118.4 ng m$^{-3}$). These concentrations are similar to those observed in Beijing during summer
2017 (237.1 ng m$^{-3}$, (Bryant et al., 2020)), but higher than those observed in Shanghai in 2018 (40.4
ng m$^{-3}$) and 2019 (34.3 ng m$^{-3}$) (Wang et al., 2021b). As previously discussed, OSi species have been
shown to form via the gas-phase photo-oxidation of isoprene, with the reactive update of the
oxidised species into to particulate phase via sulfate (Lin et al., 2013a; Surratt et al., 2010). Recently,
a heterogeneous photo-oxidation pathway from 2-MT-OS (C$_5$H$_{12}$O$_7$S) to several OSi species was
proposed, including C$_5$H$_{10}$O$_7$S, C$_5$H$_8$O$_7$S, C$_5$H$_{12}$O$_8$S, C$_5$H$_{10}$O$_8$S and C$_4$H$_8$O$_7$S (Chen et al., 2020). 2-MT-OS
showed moderate correlations (pre-monsoon : R$^2$ = 0.52-0.72, post-monsoon: R$^2$ = 0.14-0.35) with
these OSi tracers that were lower than observed in Beijing summer (R$^2$ = 0.83-0.92) (Bryant et al.,
2020). These correlations could suggest that this is a more common formation route in pre-monsoon
Delhi, than in post-monsoon. However, the correlations could also be driven by the common
pathways between the OSi species, with the reactive uptake of gas phase intermediates via sulfate
reactions. The lower correlations during the post-monsoon could be due to increased influences of
anthropogenic sources coupled to the stagnant conditions.



Figure 6 shows the binned OSi concentrations for each filter collection time across the pre- and post-
monsoon campaigns to create a partial diurnal profile. During the pre-monsoon, the daily variation
in OSi concentrations was much clearer, with day-time maxima and nocturnal minima, which are in
line with daily peak isoprene (Figure 3) and OH radical concentrations. The highest observed OSi
concentrations during the pre-monsoon were ~ 600 ng m$^{-3}$, which occurred at the start of the
campaign. High isoprene concentrations may have been the cause, but unfortunately isoprene
measurements were not available during this period to confirm. However, high OSi concentrations
also occurred when particulate inorganic sulfate concentrations were at their highest (Figure S6),
while sulfate measured via the HR-AMS was also high during this period (Figure 1). During the post-
monsoon, although a similar diurnal pattern was observed, the variation was less pronounced, with
higher OSi concentrations observed at the start and end of the campaign (Figure 5). The low OSi
concentrations during the middle of the campaign, coincide with lower isoprene and inorganic
sulfate concentrations, but also low VC values, suggesting more stagnant conditions.
The sum of OSi species across all filters sampled showed a variable correlation with particulate
sulfate across both campaigns. The pre-monsoon correlation was similar to those observed in
Beijing, Guangzhou and the SE-US ($R^2$: 0.55)(Bryant et al., 2020, 2021; Budisulistiorini et al., 2015;
Rattanavaraha et al., 2016a) while the post-monsoon was significantly weaker ($R^2$: 0.28). However, a
clear relationship between OSi tracers and inorganic sulfate can be seen in Figure 7 across both
campaigns, where the highest OSi concentrations occurred under the highest $SO_4^{2-}$ concentrations.
During the post-monsoon campaign, OSi concentrations levelled off at high sulfate concentrations.
In the pre-monsoon this levelling off is not observed, potentially due to the lower number of
samples. The high concentrations of organics measured by the HR-AMS (Table S2) during the post-
monsoon (48.7 ± 35.4) μg m$^{-3}$ compared to the pre-monsoon (19.8 ± 13.7) μg m$^{-3}$, suggests the
reactive uptake of the gaseous OSi intermediates to the aerosol phase may be limited due to
extensive organic coatings on the sulfate aerosol. Multiple studies have now shown that organic
coatings on sulfate aerosol can limit the reactive uptake of IEPOX, suggesting the pre-monsoon is
volume limited but the post-monsoon is diffusion limited. (Gaston et al., 2014; Lin et al., 2014; Riva
et al., 2016c)
Isoprene NOS (NOSi) have been shown to be produced by photo-oxidation in the presence of NO
and from $NO_3$ oxidation chemistry (Hamilton et al., 2021; Ng et al., 2017; Surratt et al., 2008b). Ten
different NOSi tracers were screened for across the two campaigns, with eight identified in the pre-
monsoon and ten in the post-monsoon. These tracers included: mono-nitrated ($C_5H_9O_{10}NS$,
$C_5H_{11}O_9NS$, $C_5H_{11}O_8NS$), di-nitrated ($C_5H_{10}O_{11}N_2S$), and tri-nitrated ($C_5H_9O_{13}N_3S$) species. These tracers
have been identified previously in China (Bryant et al., 2020, 2021; Hamilton et al., 2021; Wang et
al., 2018b, 2021b). Unlike the OSi tracers, total NOSi concentrations were on average higher during
the pre-monsoon (32.6 ± 19.9) ng m$^{-3}$ compared to the post-monsoon (20.2 ± 13.3) ng m$^{-3}$. This is
likely due to extremely high night-time NO concentrations during the post-monsoon quenching $NO_3$
radicals, limiting the isoprene + $NO_3$ pathway. The NOSi time series and diurnal shown in Figures 5
and 6 respectively highlight the strong nocturnal enhancements in concentrations during the pre-
monsoon, suggesting isoprene + $NO_3$ formation pathway is dominant. Due to the long sampling time,
it is likely that these species are forming in the early evening as $NO_3$ oxidation becomes more
competitive with OH, while isoprene concentrations are still relatively high. During the post-
monsoon, NOSi concentrations were highest at night and the early morning. The high morning
concentrations could be due to non-local sources mixing down as the shallow night-time boundary
layer breaks down. Ideally, future work in Delhi or India should focus on the measurements of
radicals and OH reactivity ($k_{OH}$), in order to improve our understanding of the chemistry occurring in
extremely polluted environments. A large spike in $NOS_i$ concentrations is observed at the start of the



post-monsoon campaign, which was not observed for the $OS_i$ tracers, this coincides with lower NO
concentrations than the rest of the post-monsoon campaign, reducing the $NO_3$ quenching by NO,
allowing for more isoprene + $NO_3$ oxidation. The $NOS_i$ species did not correlate towards particulate
sulfate ($R^2 < 0.2$) across either campaign, suggesting that uptake onto sulfate is not the limiting step
in $NOS_i$ formation (unlike for the $OS_i$ species).




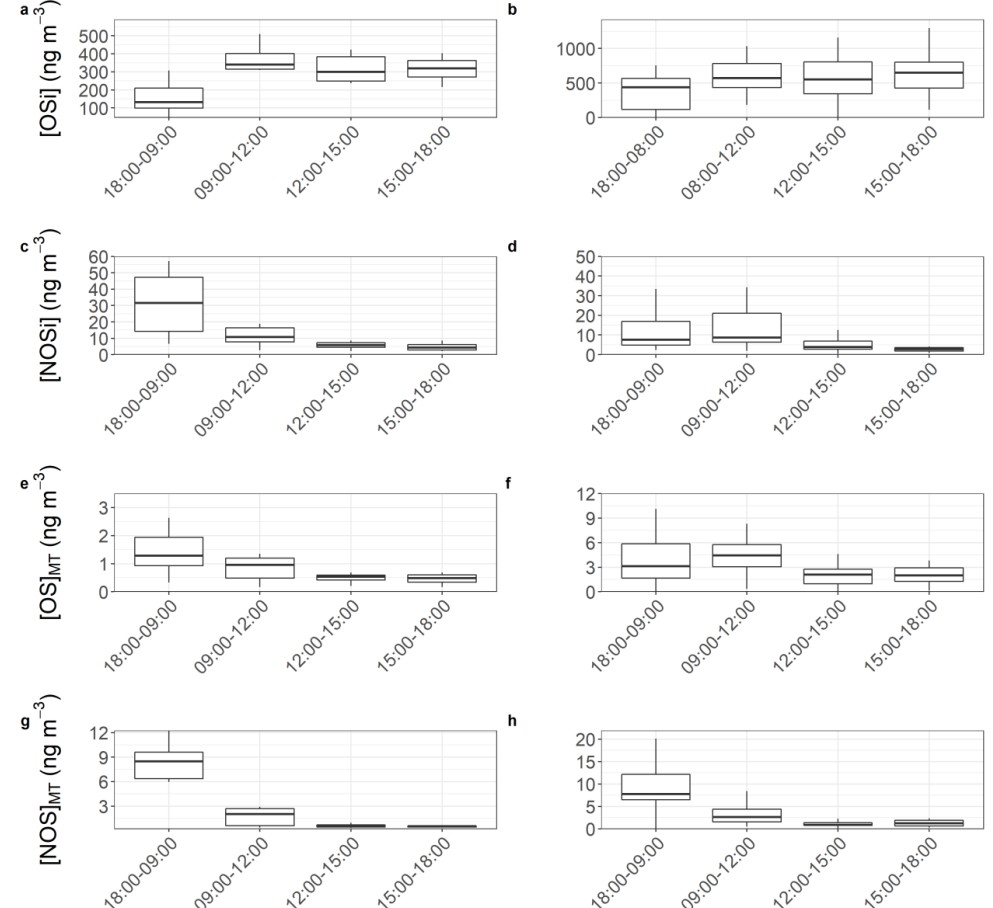

Filter collection time



Figure 6. Partial diurnal variations from the binned concentrations of $OS_i$, $NOS_i$, $OS_{MT}$ and $NOS_{MT}$ concentrations at each filter collection time across the pre (left) and post-monsoon (right) campaigns. The lower and upper part of the box representing the 25[th] and 75[th] percentiles, with the upper and lower lines extending no further than 1.5 times the interquartile range of the highest and lowest values within the hinge respectively. Only species identified in more than 40 % of the samples for each campaign were included.

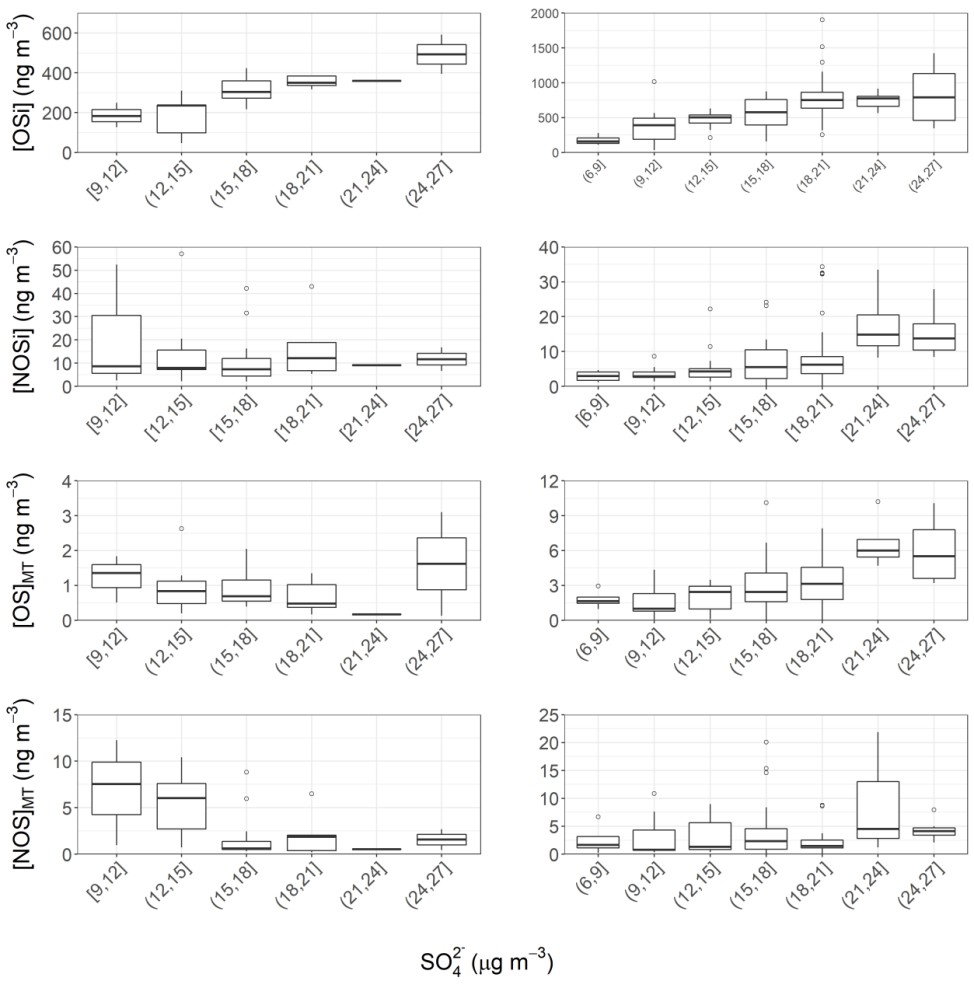



Figure 7. Quantified SOA (OS$_i$, NOS$_i$, OS$_{MT}$, NOS$_{MT}$) vs inorganic sulfate concentrations across the
pre- (left) and post-monsoon (right) campaigns. The lower and upper part of the box
representing the 25$^{th}$ and 75$^{th}$ percentiles, with the upper and lower lines extending no further
than 1.5 times the interquartile range of the highest and lowest values within the hinge
respectively. Only species identified in more than 40 % of the samples for each campaign were
included.







**3.8 Monoterpene secondary organic aerosol**


Monoterpene derived OS ($OS_{MT}$) and NOS ($NOS_{MT}$) markers have also been identified from the
oxidation by OH, $NO_3$ and $O_3$ in the presence of $SO_2$ or sulfate seed in simulation studies
(Brüggemann et al., 2020b; Iinuma et al., 2007; Kleindienst et al., 2006; Surratt et al., 2008a; Zhao et
al., 2018). Compared to isoprene, the ozonolysis of monoterpenes is a key degradation pathway,
with higher SOA yields from ozonolysis observed when compared to isoprene (Åsa M. Jonsson et al.,
2005; Atkinson and Arey, 2003; Eddingsaas et al., 2012a, 2012b; Kristensen et al., 2013; Mutzel et
al., 2016; Simon et al., 2020; Zhao et al., 2015). A recent study in the SE-US suggests that the
degradation of 80 % of monoterpenes at night is due to ozonolysis at that location (Zhang et al.,
2018). Monoterpene derived OS and NOS species have been extensively observed, with ON
contributing considerably to OA (Lee et al., 2016; Xu et al., 2015; Zhang et al., 2018). Recently NOS
hydrolysis has also been shown to be a potential formation route of OS particle phase species (Darer
et al., 2011; Passananti et al., 2016).
Twenty-three monoterpene-derived organosulfate ($OS_{MT}$) species, which have been seen previously
in chamber (Surratt et al., 2008b) and ambient studies (Brüggemann et al., 2019; Wang et al., 2018b,
2021b), were identified across the pre- and post-monsoon campaigns. It should be noted that
recently $OS_{MT}$ artefacts has been shown to form when filters have been sampled without a denuder.
(Brüggemann et al., 2020b). However, the strong diurnal variations of the $OS_{MT}$ species, and lack of
correlation with $SO_2$ suggest this process is unlikely to have contributed significantly to the $OS_{MT}$
measured in this study. Post-monsoon concentrations were similar (3.96 ± 1.6) ng m$^{-3}$ to the pre-
monsoon (3.05 ± 1.3) ng m$^{-3}$, with $C_9H_{16}O_6S$ the dominant species across both campaigns,
contributing on average ~ 29 % of the $OS_{MT}$ mass. It should be noted that the majority of the $OS_{MT}$
were not identified in every sample, and as such only tracers which were identified in at least 40 %
of the samples were examined further.
Total $OS_{MT}$ showed a strong diurnal profile across both campaigns, peaking at night, with an
afternoon minimum (Figures 5 & 6). During the pre-monsoon campaign, the highest $OS_{MT}$
concentrations were observed during a day-time sample, coinciding with peak sulfate and NO
concentrations. Both limonene and α-pinene also show peaks during this filter sampling period of ~
0.05 ppbv. Spikes in limonene and α-pinene concentrations were also observed on the 31$^{st}$ of May,
but $OS_{MT}$ concentrations were much lower, likely due to the lower sulfate concentrations. During the
post-monsoon campaign, nocturnal enhancements are observed (Figure 6), suggesting MT + $NO_3$
chemistry is important. Like the NOSi markers, higher $OS_{MT}$ concentrations were observed during the
early morning sample, likely due to a lower PBLH concentrating the markers coupled to MT+OH/$O_3$
occurring after sunrise in the post-monsoon. The night-time formation of the $OS_{MT}$ species is in line
with previous studies (Bryant et al., 2021), and with the diurnal variations of α-pinene and limonene,
which peak at night. Previous chamber studies investigating reactions of monoterpenes with $NO_3$
radicals have also shown formation of $OS_{MT}$ with the same molecular formulae as measured here
(Surratt et al., 2008a).
$OS_{MT}$ concentrations observed in Delhi are much lower than those of the OSi, similar to other studies
(Hettiyadura et al., 2019; Wang et al., 2018b, 2021b). Considering the high concentrations of
extremely reactive α-pinene and limonene observed during the post-monsoon period, higher $OS_{MT}$
concentrations might be expected. One possible reason for the low $OS_{MT}$ is the inability of $OS_{MT}$
precursor species to undergo reactive uptake into the aerosol phase under atmospherically relevant
acidic conditions, with chamber studies suggesting extremely acidic conditions are needed for



uptake to occur (Drozd et al., 2013). Delhi is characterised by large concentrations of free ammonia
and alkaline dust, and previous studies have highlighted that it has less acidic aerosol (pH 5.7 – 6.7,
Kumar et al., 2018) across the year than Beijing (pH 3.8 – 4.5, Ding et al., 2019) and the SE-US (pH 1.6
– 1.9, Rattanavaraha et al., 2016a).
Unlike the $OS_{MT}$ species, the $NOS_{MT}$ species ($C_{10}H_{17}NO_7S$, $C_9H_{15}NO_8S$, $C_{10}H_{17}NO_9S$, $C_9H_{15}NO_9S$,
$C_{10}H_{17}NO_8S$) showed strong seasonality, with pre- and post-monsoon concentrations of (7.6 ± 3.8) ng
m$^{-3}$ and (17.6 ± 6.1) ng m$^{-3}$ respectively. This is opposite to the quantified NOSi species, which
showed higher pre-monsoon concentrations. This is likely due to much higher post-monsoon
concentrations of monoterpenes. Of the $NOS_{MT}$ species observed, $C_{10}H_{17}NO_7S$ was the most
abundant, contributing on average 79 % and 76 % of the $NOS_{MT}$ concentrations across the pre- and
post-monsoon respectively. Previous studies have also highlighted $C_{10}H_{17}NO_7S$ to be the dominant
monoterpene derived sulfate containing tracer (Wang et al., 2018b). In the post-monsoon nine
$C_{10}H_{17}NO_7S$ isomers were observed, and seven in the pre-monsoon. The summed $C_{10}H_{17}NO_7S$
concentrations during the pre- (5.96 ± 3.33) ng m$^{-3}$ and post-monsoon (13.36 ± 4.98) ng m$^{-3}$, are of a
similar magnitude to those observed in other locations as shown in Table 2.  These concentrations
are also similar to those quantified by authentic standards across four Chinese megacities (Wang et
al., 2021a). Like the $OS_{MT}$ species, some $NOS_{MT}$ species were not identified in many of the filter
samples, and as such tracers which were observed in more than 40 % of the samples were summed
for further analysis. The $NOS_{MT}$ pre-monsoon time series (Figure 5) shows a similar temporal profile
to the NOSi species, with lower concentrations during the enhancement in NO concentrations
(Figure S4) at the start of the campaign. $NOS_{MT}$ showed strong diurnal variations across both
campaigns (Figure 6), peaking at night with lower concentrations during the afternoon, as seen
previously (Bryant et al., 2021; Wang et al., 2018b). Therefore, the formation of $NOS_{MT}$ is likely
dominated by $NO_3$ radical chemistry.  Both $NOS_{MT}$ and $OS_{MT}$ species showed limited correlation
towards $SO_2$ and particulate sulfate ($R^2 < 0.1$), indicating that although sulfate is essential to their
formation, sulfate availability does not control $NOS_{MT}$ concentrations.

### 628  3.10 Contributions of total quantified SOA (qSOA) to particulate mass

Particulate concentrations in Delhi are among the highest across the world (WHO, 2018), with
concentrations over 600 μg m$^{-3}$ being observed during this study. qSOA, defined here as the sum of
all OSi, NOSi, $OS_{MT}$, and $NOS_{MT}$ tracers quantified (including those not identified in more than 40 % of
the samples), was calculated to determine the total contribution these species make to particulate
mass in Delhi.  Total oxidised organic aerosol (OOA), a proxy for SOA in PM$_1$, was derived from the
HR-AMS measurements during the pre- and post-monsoon campaigns, with averages of (19.8 ± 13.7)
μg m$^{-3}$ and (48.7 ± 35.4) μg m$^{-3}$ respectively. qSOA contributed on average (2.0 ± 0.9) % and (1.8 ±
1.4) % to the total OOA. Isoprene and monoterpene derived species contributed on average 83.2 %
and 16.8 % of qSOA across the pre-monsoon respectively compared to 81.5 % and 18.5 % during the
post-monsoon respectively. During certain periods qSOA contributed a maximum of 4.2 % and 6.6 %
to OOA during the pre- and post-monsoon, respectively. This is under the assumption that when the
OS and NOS species fragment in the AMS ion source they lose their sulfate and nitrate groups. This is
similar to the contributions made by OSi markers in Beijing to total OOA (2.2 %) (Bryant et al., 2020).
Previous studies in the SE-US have reported much higher contributions of isoprene species to total
OA. As quantified by an aerosol chemical speciation monitor, summed iSOA tracers on average
accounted for  9.4 % of measured OA at Look Rock, downwind of Maryville and Knoxville, but up to a
maximum of 28.1 % (Budisulistiorini et al., 2015). This is lower than that measured at a rural site at



Yorkville, Georgia with just low-NO isoprene SOA tracers accounting for between 12-19 % of total OA
(Lin et al., 2013b).
Sulfate was also measured in the $PM_1$ size range by HR-AMS, with pre- and post-monsoon mean
concentrations of (7.5 ± 1.8) µg m$^{-3}$ and (5.5 ± 2.7) µg m$^{-3}$. The sulfate containing OS and NOS species
quantified in this study may fragment in the AMS to produce a sulfate signal which is not related to
inorganic sulfate. To estimate the contribution that sulfate contained within qSOA species could
make to total AMS sulfate, the quantified mass of sulfate contained within each marker was
calculated based on the fraction of sulfate to each marker molecular mass. For example, 2-MT-OS
has an accurate mass of $m/z$ 216.21, meaning the percentage of 2-MT-OS mass associated with
sulfate is ~44 %. During the pre-monsoon campaign the qSOA sulfate accounted for on average 2.2
% to the total $PM_1$ sulfate, but up to 4.8 % on certain days. qSOA contributed considerably more to
the sulfate in the post-monsoon campaign, with an average of (6.1 ± 4.5) % with a maximum of 18.7
%. This finding indicates the need to consider the sources of particulate sulfate measured by the
AMS when calculating aerosol pH. The sulfate contribution from the fragmentation of common small
OS compounds (hydroxymethylsulfonate, methylsulfonic acid) can be distinguished in the AMS using
the relative ratio of sulfur containing peaks.(Chen et al., 2019; Javed et al., 2021) However, more
work is needed to determine how larger OS and NOS fragment in the AMS such as those quantified
in this study. Overall, this highlights that isoprene and MT oxidation can make significant
contributions to organic and sulfate-containing aerosol, even in extremely polluted environments
such as Delhi. It should be noted that this is just a subset of potentially many more SOA from
isoprene and monoterpene markers and only focusses on sulfate containing species.

**Conclusion**

Isoprene- and monoterpene-derived organosulfate (OS) and nitrooxy organosulfate (NOS) species
were quantified during pre- and post-monsoon measurement periods in the Indian megacity of
Delhi. An extensive dataset of supplementary measurements was obtained alongside filter samples,
including isoprene and speciated monoterpenes. Isoprene and monoterpene emissions were found
to be highly influenced by anthropogenic sources, with strong correlations to anthropogenic tracers
at night across both campaigns. High nocturnal concentrations of pollutants were observed due to a
low boundary layer height and stagnant conditions, especially during the post-monsoon period.
Isoprene OS markers (OSi) were observed in higher concentrations during the post-monsoon (557 ±
423) ng m$^{-3}$ compared to the pre-monsoon campaign (238 ± 118) ng m$^{-3}$. OSi showed a moderate
correlation with inorganic sulfate across both campaigns. However, concentrations levelled off at
high sulfate concentrations during the post-monsoon which is consistent with organic coatings
limiting uptake of isoprene epoxides.  Isoprene NOS species (NOSi) showed nocturnal enhancements
across both campaigns, while the highest average concentrations were observed in the morning
samples of the post-monsoon campaign. The high morning concentrations are likely due to the
oxidation of VOCs by OH radicals from photolytic processes throughout the morning. Monoterpene
derived OS ($OS_{MT}$) and NOS ($NOS_{MT}$) markers were observed to have nocturnal enhancements in
concentrations, in-line with their precursors. $NOS_{MT}$ markers were observed in similar concentrations
to those of other megacities. Total quantified SOA contributed on average (2.0 ± 0.9) % and (1.8 ±
1.4) % to the total OOA. Considering high OOA concentrations were observed across the two
campaigns, the total markers contributed up to a maximum of 4.2 % and 6.6 % across the pre- and
post-monsoon respectively. Overall, this work highlights that even small numbers of isoprene and





monoterpene derived SOA markers can make significant contributions to OA mass, even in highly
polluted megacities.

Table 1. Molecular formulae, retention times and time weighted means (ng m$^{-3}$) of
organosulfates (OS) and nitrooxy oganosulfates (NOS) from isoprene (i) and monoterpenes
(MT) observed across pre- and post-monsoon campaigns in Delhi.

| Class | Molecular formula | Pre- | SD | Post- | SD | RT's (min) |
|---|---|---|---|---|---|---|
| OS$_i$ | $C_5H_{12}O_7S$ | 38.79 | 30.19 | 17.91 | 19.87 | 0.71 |
| | $C_5H_{10}O_5S$ | 26.16 | 23.30 | 53.63 | 131.19 | 0.93 |
| | $C_2H_4O_6S$ | 21.35 | 18.27 | 84.65 | 82.79 | 0.73 |
| | $C_5H_{10}O_6S$ | 19.80 | 13.78 | 45.87 | 29.47 | 0.79 |
| | $C_4H_8O_7S$ | 19.70 | 12.48 | 47.96 | 39.01 | 0.73 |
| | $C_3H_6O_5S$ | 19.50 | 12.47 | 35.27 | 40.15 | 0.73 |
| | $C_5H_8O_7S$ | 18.76 | 11.01 | 38.75 | 25.34 | 0.73 |
| | $C_4H_8O_6S$ | 16.57 | 9.77 | 45.48 | 37.46 | 0.74 |
| | $C_5H_{10}O_7S$ | 11.82 | 7.04 | 25.89 | 18.06 | 0.73 |
| | $C_3H_6O_6S$ | 6.64 | 5.00 | 38.06 | 40.30 | 0.73 |
| | $C_4H_8O_5S$ | 6.46 | 4.08 | 22.44 | 21.39 | 0.75 |
| | $C_5H_{10}O_8S$ | 6.25 | 5.07 | 7.00 | 5.54 | 0.73 |
| | $C_2H_4O_5S$ | 5.33 | 3.37 | 15.92 | 13.79 | 0.73 |
| | $C_2H_6O_5S$ | 5.23 | 6.36 | 24.99 | 20.38 | 0.73 |
| | $C_5H_8O_5S$ | 5.16 | 2.57 | 7.87 | 7.93 | 0.85 |
| | $C_3H_6O_7S$ | 3.54 | 3.49 | 14.78 | 11.50 | 0.75 |
| | $C_5H_{12}O_6S$ | 2.01 | 1.23 | 6.53 | 4.32 | 0.74 |
| | $C_3H_8O_6S$ | 1.90 | 1.08 | 12.25 | 10.82 | 0.75 |
| | $C_5H_8O_9S$ | 1.20 | 1.04 | 2.12 | 1.85 | 0.64 |
| | $C_4H_6O_6S$ | 1.10 | 0.76 | 8.61 | 15.65 | 0.74 |
| | $C_5H_{12}O_8S$ | 0.55 | 0.43 | 0.65 | 0.61 | 0.75 |
| | Total | 237.83 | | 556.64 | | |
| NOS$_i$ | $C_5H_{10}O_{11}N_2S$ | 18.65 | 8.77 | 11.63 | 8.09 | 1.39,1.92,2.85,3.4 |
| | $C_5H_{11}O_9NS$ | 8.55 | 5.71 | 5.93 | 5.06 | 0.86 |
| | $C_5H_9O_{10}NS$ | 3.91 | 3.46 | 1.42 | 1.31 | 0.94 |
| | $C_5H_{11}O_8NS$ | 1.52 | 0.84 | 1.17 | 1.20 | 1.09 |
| | $C_5H_9O_{13}N_3S$ | 0.002 | 0.001 | 0.011 | 0.009 | 6.67,7.89,8.06 |
| | Total | 32.63 | | 20.15 | | |
| OS$_{MT}$ | $C_9H_{16}O_6S$ | 1.10 | 0.61 | 1.67 | 0.88 | 6.67/7.14/7.5/8.3 |
| | $C_{10}H_{18}O_5S$ | 0.56 | 0.63 | 0.10 | 0.12 | 3.39 |
| | $C_{10}H_{16}O_5S$ | 0.28 | 0.13 | 0.77 | 0.06 | 4.91/7/9.08/10.9/11.33/11.97/13.26 |
| | $C_{10}H_{20}O_7S$ | 0.25 | 0.21 | 0.27 | 0.21 | 4.19 |
| | $C_{10}H_{16}O_7S$ | 0.23 | 0.15 | 0.21 | 0.13 | 3.61/11.68 |
| | $C_9H_{16}O_7S$ | 0.16 | 0.17 | 0.22 | 0.19 | 4.39/6.77 |
| | $C_{10}H_{18}O_6S$ | 0.15 | 0.10 | NA | NA | 10.27 |



| | | | | | |
|---|---|---|---|---|---|
| $C_9H_{14}O_6S$ | 0.15 | 1.10 | 0.25 | 0.14 | 3.5/5.81 |
| $C_{10}H_{16}O_6S$ | 0.10 | 0.06 | 0.06 | 0.03 | 9.33 |
| $C_{10}H_{18}O_8S$ | 0.02 | 0.01 | 0.04 | 0.24 | 7.24 |
| $C_8H_{14}O_7S$ | 0.04 | 0.03 | 0.10 | 0.15 | 4.46 |
| Total | 3.05 | | 3.68 | | |
| $C_{10}H_{17}NO_7S$ | 5.96 | 3.33 | 13.36 | 4.98 | 9.1/10.16/10.67/10.92/11.07/11.36/11.57/12.01/13.28 |
| $C_9H_{15}NO_8S$ | 1.12 | 0.51 | 2.79 | 1.14 | 3.5/5.81 |
| $C_{10}H_{17}NO_9S$ | 0.47 | 0.19 | 1.15 | 0.29 | 3.93/5.34/6.39/7.89/9.26/10.11/17.94 |
| $C_9H_{15}NO_9S$ | 0.0216 | 0.0044 | 0.22 | 0.14 | 2.69/3.46 |
| $C_{10}H_{17}NO_8S$ | 0.01 | 0.01 | 0.07 | 0.04 | 5.77 |
| Total | 7.59 | | 17.59 | | |

(Row group label: $NOS_{MT}$)






Table 2. Comparison of $C_{10}H_{17}NO_7S$ concentrations across different locations.
Locations and concentrations in bold were quantfified by authentic standards.

| Location | $C_{10}H_{17}NO_7S$ (ng m$^{-3}$) | Reference |
|---|---|---|
| **Delhi Pre-monsoon** | **5.96** | **This study** |
| **Delhi Post-monsoon** | **13.36** | **This study** |
| Guangzhou summer | 7.15 | Bryant et al., 2021 |
| Guangzhou winter | 11.11 | Bryant et al., 2021 |
| Shanghai 15/16 | 6.21 | Wang et al., 2021b |
| Shanghai 16/17 | 5.55 | Wang et al., 2021b |
| Beijing | 12.00 | Wang et al., 2018b |
| Atlanta | 9.00 | Hettiyadura et al., 2019 |
| Hong Kong | 5.61 | Wang et al., 2021a |
| Guangzhou | 12.32 | Wang et al., 2021a |
| Shanghai | 16.51 | Wang et al., 2021a |
| Beijing | 13.15 | Wang et al., 2021a |


**Data availability**

Data used in this study can be accessed from the CEDA
archive: https://catalogue.ceda.ac.uk/uuid/ba27c1c6a03b450e9269f668566658ec (Nemitz et al.,
707 2020).

**Author contributions**

DJB prepared the manuscript with contributions from all authors. DJB, BSN, SJS, SHB, WSD, ARV,
JMC, WJFA, BL, EN and JRH provided measurements and data processing of pollutants used in this



study. MJN and ARR contributed to scientific discussion. S, RG, BRG, TH and EN assisted with
logistics. CNH, JDL, ARR, JFH provided overall guidance to the experimental setup and design.
**Acknowledgements**
The authors acknowledge Tuhin Mandal at CSIR-National Physical Laboratory for his support in
facilitating the measurement sites used in this project and Gareth Stewart for the VOC
measurements. This work was supported by the Newton Bhabha fund administered by the UK
Natural Environment Research Council through the DelhiFlux and ASAP projects of the Atmospheric
Pollution and Human Health in an Indian Megacity (APHH-India) programme. The authors gratefully
acknowledge the financial support provided by the UK Natural Environment Research Council and
the Earth System Science Organization, Ministry of Earth Sciences, Government of India, under the
Indo-UK Joint Collaboration (DelhiFlux). Daniel J. Bryant and Beth S. Nelson acknowledge the NERC
SPHERES doctoral training programme for studentships. James M. Cash is supported by a NERC E3
DTP studentship.
**Financial support**
This research has been supported by the Natural Environment Research Council (grant nos.
NE/P016502/1 and NE/P01643X/1) and the Govt. of India, Ministry of Earth Sciences (grant no.
MoES/16/19/2017/APHH, DelhiFlux).
**Competing interests**
The authors declare that they have no conflict of interest.

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
