# Peer review of "Biogenic and anthropogenic sources of isoprene and monoterpenes and their secondary organic"

_Atmospheric Chemistry and Physics, 2022_

## Author Comment (AC1)

**Biogenic and anthropogenic sources of isoprene and monoterpenes and their secondary organic aerosol in Delhi, India**

We would like to thank the reviewers for their insightful comments. Responses to the comments are given below.

**Reviewer 1.**

**One of my major concerns is that even though the title says "secondary organic aerosol", the SOA section of the paper almost entirely revolves around the NOS and OS species in SOA from isoprene and/or monoterpenes, which constitute a small fraction of the total SOA. Either the relevant discussion sections should be expanded to include some broader details of SOA from these precursors, or, the sub-section titles should be changed to more accurately represent the discussion contained therein.**

Subsection titles have been updated to better reflect the content.

"3.5 Isoprene and monoterpene OS and NOS formation"

"3.5.1 Isoprene OS and NOS markers"

"3.5.2 Monoterpene OS and NOS markers"

"3.5.2 Contributions of total Isoprene and monoterpene OS and NOS (qSOA) to particulate mass"

**Based on the title, I was very curious about the anthropogenic sources of isoprene and monoterpenes in Delhi. However, I think that potential anthropogenic contributions are not discussed sufficiently in the paper. For example, correlations with CO are briefly discussed, and biomass burning and VCPs are hinted at as likely contributors. But this is all general information and as a reader, I am not able to gain significant insights into anthropogenic sources of terpenes in Delhi. I would appreciate seeing some correlations of compounds of interest with D5 or Benzyl alcohol. D5 can be measured via GC techniques so it would be useful to include at least some information on it (e.g. temporal trends even if ion abundance-based). Delhi is densely populated and characterized by a lot of street activity with temperatures reaching 30-40 C (premonsoon) so I would assume that there should be D5 in Delhi's air.**

This is an interesting comment. Unfortunately, due to data availability we do not have any D5 or Benzyl-alcohol time series. Based on the current correlations towards CO, we observe an anthropogenic source of monoterpenes, however clear sources cannot be identified from this data, but will certainly be a key focus of future studies.

**Section 2.3: Please add some sentences about the mass resolution of the spectrometer. Also, it would be good to add a brief discussion (3-4 sentences) about the quantification method before citing Bryant et al. 2021.**

Additional text has been added:

",with a mass resolution of 140,000."

The section has had additional detail added. It now reads:

"Isoprene OS and NOS markers were quantified using authentic standards of 2-MG-OS and 2-methyl tetrol OS (2-MT-OS) with later eluting monoterpene OS and NOS quantified using camphorsulfonic acid. Standards were run across a 9-point calibration curve (2 ppm – 7.8ppb, $R^2 > 0.99$) More details about the method can be found in Bryant et al., 2021."

**Figure 3 (e,f): a-pinene concentrations do not show significant dilution during premonsoon daytime conditions when compared to post-monsoon. This is despite the daytime PBLH being substantially higher during pre-monsoon than post. Shouldn't your discussion in lines 405-408 apply here? Is there an explanation for this?**

As shown in Figure S2, the pre-monsoon PBLH diurnal highlights the difference in day time ( ̃ 2000 m) and night time (500 m) heights are around a factor of 4. This is compared to the post-monsoon PBLH, with a daytime ( ~1500 m) and night-time ( ~100 m) height difference of a factor of ~15. As such, the PBLH effect is much stronger during the post-monsoon, hence the stronger diurnal variations compared to the pre-monsoon. Overall the PBLH effect is weaker during the pre-monsoon compared to the post-monsoon.

Additional text has been added: "The diurnal variations across both campaigns are likely driven by both emissions as well as boundary layer effects. The boundary layer effect however is much stronger during the post-monsoon, with a shallower nocturnal boundary layer, as such the post-monsoon period has a more pronounced diurnal."

**Lines 497-499: Shouldn't low ventilation and stagnant conditions lead to greater accumulation and higher concentrations of isoprene and sulfate? Or the magnitude of sources also drops?**

The boundary layer effects on isoprene are explained across the lines 300-320 : "Figure 3 indicates that average daytime peak isoprene mixing ratios during the pre-monsoon campaign were roughly double that of the post-monsoon campaign. In contrast, average nocturnal mixing ratios of isoprene were 5 times higher in the post-monsoon compared to the pre-monsoon ((0.65 ± 0.43) ppbv versus (0.13 ± 0.18) ppbv). In the post-monsoon campaign, isoprene mixing ratios show a strong biogenic emission driven diurnal profile at the start of the campaign. However, towards the end of the post monsoon measurement period, the isoprene mixing ratios become less variable with a high mixing ratio maintained overnight (Figure 2). This is potentially due to more stagnant conditions as observed by the VC in Figure S1."

We believe that due to sulfate being a secondary pollutant rather than an emission, the diurnal is more influenced by longer range transport and is therefore not influenced as strongly by boundary layer conditions.

Additional text has been added: "Due to the secondary nature of sulfate, the sulfate concentrations are less likely to be influenced by the boundary layer effects, compared to directly emitted VOCs."

**Line 505: Please be consistent in using "sulfate" versus its formula in the text.**

"SO42-" has been removed from text and replaced with "sulfate".

**Line 577: Adding a brief discussion on C9H16O6S (or citations) would be of help hereto an unfamiliar reader, especially since it contributes a large fraction to the OS(MT) mass. Is this species consistently observed in OS(MT) across different sites?**

Additional text and references have been added:

"$C_9H_{16}O_6S$ has been observed in chamber studies (Surratt et al., 2008a) as well as in ambient samples in Denmark, Shanghai and Guangzhou previously(Bryant et al., 2021; Nguyen et al., 2014; Wang et al., 2017)."

**Lines 582-587: The authors should discuss what changed between pre- and postmonsoon around the site that led to seasonal variation in the significance of atmospheric reaction chemistry (e.g. MT+NO3 being more important in post-monsoon).**

Unfortunately, we do not have enough data to say with certainty if the emissions increased between the pre and post-monsoon. We believe that the difference in concentrations is likely driven by boundary layer conditions, with lower PBLH during both the day and night during the post-monsoon compared to the pre-monsoon. Emission sources such as traffic and cooking are unlikely to have dramatically changed between seasons. Additional biomass burning within the city for heating may be an additional source during the colder post-monsoon period and significant seasonal regional agricultural burn off of crop residues.

**Lines 636-638: This sentence is confusing. qSOA is all isoprene and monoterpene derived species. So how come the fractions are not all 100%?**

We are not sure what the reviewer is asking here. As stated in the text, isoprene and monoterpene species contributed 83.2 % and 16 .8 % to qSOA (83.2 + 16.8 = 100) during the premonsoon. And 81.5 % and 18.5 % (81.5 + 18.5 = 100) during the post monsoon.

**Line 608: Are the higher post-monsoon concentrations of monoterpenes only due to lower PBLH or are the source profiles any different?**

Unfortunately, we cannot say for certain. There are likely multiple factors leading to the higher post-monsoon concentrations. This is likely due to an accumulation of species due to boundary layer conditions, which quenches the nocturnal chemistry meaning the concentrations don't deplete, as well as increased emissions due to the time of year, with an increase in biomass burning. (https://aaqr.org/articles/aaqr-13-01-oa-0031) These species however come from a range of sources, BB, traffic, cooking, and personal care products which are likely to have different seasonal emission factors.

Additional text has been added to the "Isoprene and monoterpene measurements" section:

"There are likely multiple factors leading the higher concentrations during the post-monsoon, including accumulation due to boundary layer effects, a lack of nocturnal radical chemistry and an increase in biomass burning (Jain et al., 2014)."

**Line 643: There should be some discussion in the methods section on how were the iSOA tracers quantified using an ACSM.**

This sentence is regarding instrumentation used and data taken from a measurement campaign in the US, as such is appropriately cited in the associated reference.

**Please proofread the manuscript for typos (e.g. line 434: "tarcers"; line 476: "update"; line 477: "into to").**

These typos have been addressed.

The manuscript has been further proofread.

Reviewer 2

**Line 35 – 36. The statement "this is one of the first observations in Asia, suggesting monoterpenes are dominated by anthropogenic sources" should be refined or removed, since work by Stewart et al and Nelson et al. already detail these observations for Delhi. This statement doesn't seem necessary given that the main focus of the manuscript is on isoprene and monoterpene SOA markers.**

We agree with the reviewer and have adjusted this comment accordingly.

**Lines 229- 246 and Fig 1. These mixing ratios (and the differences across seasons) are impressive, but it's difficult to see the details in the seasonal patterns in just a time series. It would be very helpful to see a third column where these gas-phase measurements are presented as diurnal patterns in order to see the seasonal and hourly differences. I would recommend this for Fig S1 as well to give the reader a better visual reference for how the meteorology impacts these mixing ratios. The authors provide a very nice discussion of the meteorological impacts in section 3.1, but I believe overlaying the diurnal patterns seasons would be very helpful.**

We chose not to combine the time series and diurnals into one plot to due the size of the figure. The diurnal plots could not be overlaid due to the difference in concentrations being too large for a single axis. The diurnals are provided in the supplementary and referenced appropriately.

**Lines 383 – 408 : The results and discussion about monoterpenes presented here reiterate many of the conclusions drawn by Stewart et al. 2021 and Nelson et al. 2021 (i.e., abundant anthropogenic source of monoterpenes). Are there other insights that can be drawn from the monoterpene data presented here? It would be helpful to see how the monoterpene distribution changes between by season or time of day (perhaps a pie chart of nighttime mixing ratios). Figure 2 suggests that alpha-pinene is the dominant monoterpene observed during the pre-monsoon season, while limonene seems to dominate during post-monsoon season. Does this point to a specific source in Delhi? Limonene is a key component of fragrances (Gkatzelis et al. 2021, Coggon et al. 2021, Peng et al. 2022) and cooking spices (Klein et al. 2016), and could be a component of biomass burning. Does this differ from the expected distribution of biogenic monoterpenes from the vegetation in Delhi?**

The average composition of monoterpenes across daytime and nighttime samples in each campaign looks  broadly similar. However, there are clear differences between the pre and post monsoon campaigns, with much higher contributions from limonene and b-ocimene. As such, we have chosen to include the comparison across the seasons, rather than day/night. Additional text has been added to the main text, as well as the following figure. Note Figure numbering has been updated accordingly.

"The average isomeric speciation of the measured monoterpenes showed low variability between day and night-time samples during each campaign, but significant differences were observed between the campaigns (Figure 4). Higher contributions from limonene and β-ocimene were

observed during the post-monsoon compared to the pre-monsoon. The reason for the difference in composition is likely due to differences in sources and/or sinks between the two periods."

[Figure]

Figure 4. Average composition of monoterpenes across the pre-monsoon and post-monsoon periods.

**Line 407: Please provide references to the OH sources noted here.**

An additional reference has been added: Lelieveld et al., 2016
https://acp.copernicus.org/articles/16/12477/2016/

**Figure 5: The legend for the pie chart is very small and difficult to read. Please make this larger to help those of us with poor eyesight!**
This has been updated.

**Lines 515 – 538: The authors mention here and elsewhere the role of high NO in quenching NO3 radicals. Indeed, the NO is very high, but I don't have a sense for how this stacks up against the other species with high reactivity towards NO3. I think this discussion could benefit from a pie chart showing the distribution of NO3 reactivity based on the VOC and inorganic gas**

**measurements, but recognizing that NO3 reactivity may be missing from the measurements (e.g. Fig 4, Liebmann et al.). I believe this would help to supplement the discussion of Figures 5 and 6.**

We thank the reviewer for this comment. We are currently working on another manuscript which will focus more on the role of the (very) high levels of NO seen in the post-monsoon campaign on radical chemistry in Delhi. This will include box modelling to investigate the sources and sinks for radicals in Delhi. This requires an extensive additional piece of work and we do not think it should be included superficially here.

**Table 1: I feel like there could be very valuable information in these distributions of isoprene and monoterpene products, and specifically for the OSMT and NOSMT speciation. Have the authors considered exploring the speciation of MT SOA markers and relating these back to the monoterpene mixing ratios observed by GC?**

We thank the reviewer once again for another insightful comment.  Due to the number and similarity of $OS_{MT}$ and $NOS_{MT}$ species that have been observed across previous studies, speciation is not possible at this stage. However, this is a key focus of our work currently.

**Line 402: The reference to Coggon et al. (2018) should be updated: https://www.pnas.org/doi/10.1073/pnas.2026653118.**

This has been updated.